# Vein Wall Invasion Is a More Reliable Predictor of Oncological Outcomes than Vein-Related Margins after Pancreaticoduodenectomy for Early Stages of Pancreatic Ductal Adenocarcinoma

**DOI:** 10.3390/diagnostics13223465

**Published:** 2023-11-17

**Authors:** Manish Ahuja, Rupaly Pandé, Shafiq Chugtai, Rachel M. Brown, Owen Cain, David C. Bartlett, Bobby V. M. Dasari, Ravi Marudanayagam, Keith J. Roberts, John Isaac, Robert P. Sutcliffe, Nikolaos Chatzizacharias

**Affiliations:** 1Department of HPB and Liver Transplant Surgery, University Hospitals Birmingham NHS Foundation Trust, Birmingham B15 2GW, UK; manish.ahuja@uhb.nhs.uk (M.A.); rupaly.pande@uhb.nhs.uk (R.P.); shafiq.chugtai@uhb.nhs.uk (S.C.); david.bartlett@uhb.nhs.uk (D.C.B.); bobby.dasari@uhb.nhs.uk (B.V.M.D.); ravi.marudanayagam@uhb.nhs.uk (R.M.); keith.roberts@uhb.nhs.uk (K.J.R.); john.isaac2@uhb.nhs.uk (J.I.); robert.sutcliffe@uhb.nhs.uk (R.P.S.); 2Department of Pathology, University Hospitals Birmingham NHS Foundation Trust, Birmingham B15 2GW, UK; rachel.brown@uhb.nhs.uk (R.M.B.);; 3Institute of Immunology and Immunotherapy, University of Birmingham, Birmingham B15 2SQ, UK

**Keywords:** portomesenteric vein groove, pancreatoduodenectomy, pancreatic ductal adenocarcinoma, survival, local recurrence, vein wall invasion, vein resection

## Abstract

Pancreaticoduodenectomy (PD) with vein resection is the only potentially curative option for patients with pancreatic ductal adenocarcinoma (PDAC) with venous involvement. The aim of our study was to assess the oncological prognostic significance of the different variables of venous involvement in patients undergoing PD for resectable and borderline-resectable with venous-only involvement (BR-V) PDAC. We performed a retrospective analysis of prospectively acquired data over a 10-year period. Of the 372 patients included, 105 (28%) required vein resection and vein wall involvement was identified in 37% of those. A multivariable analysis failed to identify the vein-related resection margins as independent predictors for OS, DFS or LR. Vein wall tumour involvement was an independent predictor of OS (risk x1.7–2) and DFS (risk x1.9–2.2) in all models, while it replaced overall surgical margin positivity as the only parameter independently predicting LR during an analysis of separate resection margins (risk x2.4). Vein wall tumour invasion may be a more reliable predictor of oncological outcomes compared to traditionally reported parameters. Future studies should focus on possible pre-operative investigations that could identify these cases and management pathways that could yield a survival benefit, such as the use of neoadjuvant treatments.

## 1. Introduction

Pancreatic ductal adenocarcinoma (PDAC) presents as a localised disease for only a small subset of patients, of whom only 20% are eligible for resection [1] with a 5-year survival rate of 6.8% [2]. Surgical treatment is the only potentially curative option and, with the advances in surgical techniques and technology, and the availability of efficacious chemotherapy regimens, the role of surgery has increased even in advanced stages of the disease [3,4,5,6], conferring survival results similar to earlier stages [7,8,9]. The results of surgical treatment are directly related to the resection margin status, with positive margins (R1) associated with early recurrence and worse overall survival (OS) [10,11]. Therefore, in the effort of performing more radical resections in order to achieve negative margins (R0), vein resections that were once considered operations of high morbidity [12] have now become the standard for PDAC with venous involvement [13,14,15,16,17,18].

Despite this, there is no universally used histopathological system in clinical practice for assessing and reporting the margins of surgical specimens, which creates difficulties in further understanding and comparing the outcomes of different studies, especially in the context of vascular resections. More specifically for pancreaticoduodenectomy (PD) specimens, a recent consensus paper [19] argued in favour of reporting the portomesenteric vein groove (PVG) surface as a margin. This is supported by the United Kingdom Royal College of Pathologists guidance, which defines the PVG as positive if cancer cells are identified within 1 mm [20], while the American College of Pathology does not follow the same recommendation [21]. The oncological significance of the PVG surface is further questioned in PD with venous resection when it is resected en bloc with the tumour. Furthermore, controversy exists as to whether vein wall tumour involvement away from the resection margins influences survival [22,23] and therefore this parameter is often not reported in the literature.

The aim of our study was to assess the oncological prognostic significance of the different variables of venous involvement in patients undergoing PD for resectable and borderline-resectable with venous-only involvement (BR-V) PDAC.

## 2. Materials and Methods

This retrospective study was conducted in a U.K. tertiary referral centre for the management of PDAC, after departmental approval, in accordance with the STROBE (Strengthening the Reporting of Observational Studies in Epidemiology) guidelines [24]. This unit adopts a policy of fast-track [25] upfront surgery approach for resectable and BR-V PDAC as supported by the UK National Institute for Care and Health Excellence [26]. Pre-operative staging investigations included a computer tomography (CT) of the thorax, abdomen and pelvis, and endoscopic ultrasound (EUS) with fine-needle aspiration (FNA) when pre-operative cytological diagnosis was required. Magnetic resonance imaging (MRI) liver and positron emission tomography (PET-CT) were used selectively if there were concerns for metastatic disease based on the CT scan. All cases were referred for adjuvant chemotherapy after the surgical treatment. The management of all cases was discussed and agreed in the hepatopancreaticobiliary multidisciplinary meeting.

All patients with PDAC in the head and the uncinate process of the pancreas who had PD (pylorus-preserving or classical) with and without venous resection were identified from a prospectively maintained database over a 10-year period (2011–2020). Patients with primary resectable and BR-V PDAC as per the National Cancer Comprehensive Guidelines (NCCN) criteria [27] were included in the study. Patients with ampullary tumours, duodenal cancer and distal cholangiocarcinoma, borderline with arterial involvement or locally advanced pancreatic tumours and patients who underwent total pancreatectomy were excluded from the study. Demographic, clinical, radiological and pathological data were obtained from the hospital’s electronic records and the departmental prospectively maintained database.

Pathological analysis of the specimens was performed in line with the Royal College of Pathologists dataset for histopathological reporting of cancers of the pancreas [20] and the TNM 8th edition was used for staging [28]. Margins were considered positive when malignant cells were identified within 1 mm of the resection margin in the paraffin-embedded specimens [20]. Vein-related surgical margins included the PVG and the venous transection margins. Furthermore, vein wall involvement, defined as involvement of the vein wall at any point of the resected vein, at or away from the margin, was also included in the analysis. The study cohort was divided into two subgroups: patients who had a vein resection and those who did not. In the first subgroup, the PVG was resected en bloc with the main specimen including part of the portal and/or superior mesenteric veins. Therefore, the PVG did not constitute a true resection margin but only an anatomical plane. In patients who did not have a vein resection, the PVG was still considered a resection margin as the portomesenteric veins were dissected from the tumour.

OS was defined as the time from diagnosis to death or last follow-up and disease-free survival (DFS) was defined as the time from resection to diagnosis of disease recurrence. In order to better account for the effects of the resection margin status, as well as separate surgical margins, survival and risk analysis were also performed separately for local recurrence (LR).

### Data Analysis

The characteristics of the cohort are presented with standard descriptive statistical analysis. Chi-square and Mann–Whitney U tests with exact statistics were used to compare nominal and ordinal variables with significance level set at *p* < 0.05. For comparison of continuous variables, one-way analysis of variance (ANOVA) was used with significance level set at *p* < 0.05. Survival analysis was performed with the Kaplan–Meier method and log-rank test was used to compare survival curves. Univariable and multivariable time-to-event analyses were performed using the Cox proportional hazard model to determine risk factors for median OS, DFS and LR. Variables were subjected to a univariable analysis first and those with *p* < 0.2 were introduced into a multivariable model. Hazard ratios (HRs) and associated 95% confidence intervals (Cis) were calculated. A two-tailed *p* value < 0.05 was considered statistically significant. All statistical analyses were performed using the software package SPSS Statistics for Windows (version 25.0; SPSS Inc., Chicago, IL, USA).

## 3. Results

A total of 372 patients were included in the study. The median age was 69 years (range 34–85) and the male to female ratio was 52%:48%. The median follow-up was 18 months (range 0–100). Approximately one third of the patients required a vein resection (28%). More vein resections were performed in female patients and, as expected, for BR-V disease. The PVG was more commonly involved in vein resections, as well as the pancreatic transection margin. Vein wall involvement was identified in 37% of patients who had a vein resection. The characteristics of the complete cohort and the subgroups are summarized in Table 1.

### 3.1. Subsection Overall Survival (OS)

There was no difference in OS between patients having PD with and without a vein resection (20 months; range 17–22 months for both groups, *p* = 0.272). Patients after R1 resections had a significantly shorter OS (18 months; range 16–20 months vs. 26 months; range: 22–31 months, *p* < 0.001). Similarly, patients with a positive PVG had a significantly shorter OS by a median of 5 months (17 months; range: 14–20 months vs. 22 months; range 19–26 months, *p* = 0.002) (Figure 1a).

Among the patients who did not have a vein resection, those with positive surgical margins had a significantly shorter OS (19 months; range 16–22 months vs. 25 months; range: 19–31 months, *p* = 0.008). Similarly, patients with a positive PVG had a significantly shorter OS by a median of 6 months (16 months; range: 12–20 months vs. 22 months; range 19–25 months, *p* = 0.044) (Figure 1b).

In the vein resection subgroup, the OS was significantly shorter in patients after R1 resection (17 months; range 12–22 months vs. 27 months; range 20–34 months, *p* = 0.001). Patients with a positive PVG also had a significantly shorter OS (19 months; range 16–22 months vs. 26 months; range 17–35 months, *p* = 0.044), while this was not observed with vein transection margins (20 months; range 16–24 months for negative vs. 17 months; range 10–24 months for positive, *p* = 0.137). The OS was significantly shorter in patients with vein wall invasion (19 months; range 14–24 months vs. 25 months; range 17–33 months, *p* = 0.014) (Figure 1c).

Table 2 presents the results of the univariable and multivariable risk analysis for OS. The multivariable analysis identified pT, pN, surgical margin status and adjuvant chemotherapy as independent predictors of OS. For patients without a vein resection, pN and adjuvant chemotherapy were the only independent predictors of OS, while for those who had a PD with vein resection the independent predictors were pT, surgical margin status, vein wall involvement and PNI. When the multivariable analysis was performed for specific resection margins, the bile duct (BD) resection margin was the only margin predicting the OS in the no vein resection subgroup, while in the vein resection subgroup only the pancreas transection margin predicted the OS. In the latter, vein wall invasion remained significant. 

### 3.2. Disease-Free Survival (DFS)

There was no difference in DFS between patients who had PD with and without a vein resection (12 months; range 11–14 months vs. 14 months; range 12–16 months, *p* = 0.120). Patients after R1 resection had a significantly shorter DFS (12 months; range 11–13 months vs. 18 months; range: 13–23 months, *p* < 0.001). Similarly, patients with a positive PVG had a significantly shorter DFS by a median of 2 months (13 months; range: 12–14 months vs. 15 months; range 12–18 months, *p* = 0.006) (Figure 2a).

Among the patients who did not have a vein resection, those after R1 resection had a significantly shorter DFS (13 months; range 11–15 months vs. 21 months; range: 16–26 months, *p* < 0.001). However, there was no difference in DFS related to the PVG status (13 months; range 11–15 months for VG positive vs. 16 months; range 13–19 months for VG negative, *p* = 0.258) (Figure 2b).

In the vein resection subgroup, the surgical margin status was significantly associated with DFS (12 months; range 10–14 months for R1 vs. 15 months; range 10–20 months for R0, *p* = 0.015). Similarly, the DFS was shorter for patients with a positive PVG (12 months; range 10–14 months vs. 13 months; range: 10–16 months, *p* = 0.029). On the contrary, vein transection margin status did not correlate with the DFS (12 months; range 2–22 months for positive vs. 13 months; range: 11–15 months for negative, *p* = 0.578). Vein wall invasion was significantly associated with the DFS (10 months; range 7–13 months for wall invasion vs. 14 months; range: 12–16 months for no invasion, *p* = 0.004) (Figure 2c).

Table 3 presents the results of the univariable and multivariable risk analysis for DFS. The multivariable analysis identified the pre-operative radiological stage, pT, pN, surgical margin status, presence of post-operative complications and adjuvant chemotherapy as independent predictors of DFS. For patients without a vein resection, the independent predictors of DFS were pT, pN, surgical margin status and adjuvant chemotherapy, while for those with vein resection only the surgical margin status and vein wall invasion were independent predictors. When a multivariable analysis was performed for specific resection margins, the only margin independently predicting DFS for the whole cohort was the SMA margin. The SMA and BD resection margins were independent predictors of DFS in the no vein resection subgroup, while in the vein resection subgroup the posterior margin was the independent predictor of DFS. In the latter, vein wall invasion remained significant.

### 3.3. Local Recurrence (LR)

There was no difference in LR between patients who had PD with and without a vein resection (28 months; range 15–41 months vs. 25 months; range 19–31 months, *p* = 0.912). Patients after R1 resection had a significantly shorter LR (20 months; range 14–26 months vs. 50 months; range: not reached, *p* < 0.001). Similarly, patients with a positive PVG had a significantly shorter LR by a median of 11 months (20 months; range: 13–27 months vs. 26 months; range 21–31 months, *p* = 0.025) (Figure 3a).

Among the patients who did not have a vein resection, the time-to-LR was significantly shorter for R1 resections (27 months; range 9–45 months vs. median not reached, *p* = 0.039). There was no significant difference observed for the PVG status as the medians were not reached in both the PVG positive and negative cases (*p* = 0.083) (Figure 3b).

In the vein resection group, the surgical margin status was significantly associated with the time-to-LR (15 months; range 1–29 months for R1 vs. median not reached for R0, *p* = 0.023). No difference was identified for the PVG status (28 months; range 12–44 months for PVG positive vs. median not reached for PVG negative, *p* = 0.364), vein transection margin status (median not reached for positive vs. 29 months; range not reached for negative, *p* = 0.307) or vein wall invasion (medians not reached, *p* = 0.191) (Figure 3c).

Table 4 presents the results of the univariable and multivariable risk analysis for LR. The multivariable analysis identified pT, pN and surgical margin status as independent predictors of LR for the whole cohort and for patients after PD without vein resection. For patients who had a PD with vein resection, the only independent predictor of LR was the surgical margin status. When multivariable analysis was performed for specific resection margins, SMA and BD resection margins were independent predictors of LR in the whole cohort and in patients who had PD without a vein resection. However, in patients who had a vein resection the only independent prognostic factor for LR was vein wall invasion.

## 4. Discussion

Disease recurrence after resection is common in PDAC and affects patient survival. Local-only recurrence accounts for up to 50% of these cases conferring survival similar to that of distant metastatic disease recurrence [10,11]. Several mitigating management strategies have been proposed including neoadjuvant treatment with chemotherapy and/or chemoradiation [29], intra-operative accentuation of the margins with irreversible electroporation [30] and intraoperative radiotherapy [31]. Variations in surgical techniques have also been described to improve margin status and hence possibly impact the rates of disease recurrence and survival [32]. Nonetheless, a significant obstacle in analysing and comparing outcomes from different studies is the lack of a common reporting system and definition of resection margins [19,21,33,34]. These differences become even more pronounced when margins relate to anatomical planes such as the PVG. The Royal Colleges of Pathologists of England [20] and Australasia [33] consider the PVG as a separate margin, even in the presence of a vein resection wherein the degree of vein invasion (tunica externa, media, intima and lumen) is reported individually. This contrasts with the American College of Pathologists, which incorporates the entire pancreatic neck as one single margin [21]. During PD, the pancreas is dissected from the portal and superior mesenteric veins, leaving the PVG to be assessed pathologically. It is not uncommon during the resection, even in resectable tumours during pre-operative imaging and in the absence of neoadjuvant treatment, for the vein to be dissected from abnormal tissue, which macroscopically is impossible to differentiate as pathological tumour involvement or stromal/fibrous reaction. In cases where this is possible, a vein resection cannot be justified due to the added complexity and possible morbidity. In cases when this dissection is not possible, a venous resection may be performed, which nowadays is considered a standard in pancreatic resectional surgery [14]. In case of a vein resection, reporting the PVG margin seems clinically irrelevant as this has been resected en bloc with the vein and therefore does not constitute a resection margin. Nonetheless, Verbeke et al. [35] reported that R0 resection can rarely be achieved in patients undergoing PD with venous resection due to the micro-anatomy of the area where there is a lack of adipose tissue between the pancreatic parenchyma and PVG.

In our study, almost a third (28%) of the 372 patients with resectable and BR-V PDAC (based on pre-operative imaging and NCCN staging criteria) had a vein resection during PD. R0 resection was reported in 46% of patients and the rates were comparable between patients requiring a vein resection or not. The different margin definitions used worldwide [36] can explain the wide range of R0 rates (29–76%) [18,37,38,39] and OS (14–35 months) reported in the literature [36,40]. As expected, the PVG was more commonly involved in cases where a vein resection was performed. The vein transection margins were involved in 16% of these cases. The pancreas transection margin was also more commonly involved after PD with vein resection, while all other specific margins rates were comparable. Tumour wall invasion was identified pathologically in 37% of cases where a venous resection was performed, while in the remaining 63% the tumour cells did not infiltrate the wall of the portal and/or superior mesenteric veins.

All survival outcomes investigated (OS, DFS and LR) were significantly longer after R0 resection, irrespective of the need for a venous resection. Positive PVG status significantly shortened the OS both after PD with (by a median of 7 months) or without (by a median of 6 months) vein resection. Interestingly, this was not the case for DFS and LR. In patients without a vein resection there was no difference in the DFS or LR related to PVG status. Nonetheless, in the vein resection subgroup, the DFS was significantly shorter for patients with a positive PVG (by a median of 1 month), but the time-to-LR was not. This is most likely explained by the fact that the majority of the disease recurrence in these patients was attributed to metastatic disease (64% of the patients with recurrence), which was also consistent with the rates of metastatic disease recurrence in the published literature [11,41]. The venous transection margins (only relevant in PD with vein resection) did not seem to affect the OS, DFS or LR, which was similar to that observed in the literature [42]; however, this may be related to the small number of patients with a positive vein transaction margin (16%). More importantly, patients with a vein wall tumour invasion (only possible to assess pathologically after PD with vein resection) had a significantly shorter OS (by a median of 6 months) and DFS (by a median of 4 months).

The results of the risk analysis for the oncological survival outcomes were also very interesting. For patients who had a PD without a vein resection, positive lymphadenopathy (stage pN1 and pN2) imposed a 4.3–5.8 times higher risk for a shorter OS, while patients who received adjuvant chemotherapy had a 4.6 times risk reduction. On the contrary, neither of these parameters were identified as independent prognostic factors of the OS in patients after a PD with vein resection. In this subgroup, pT3 increased the risk for a shorter OS by 3.7 times (Table 2). With regards to the DFS, pT3 and positive lymphadenopathy (stages pN1 and pN2) imposed a 2.1–2.4 and 3.3–4.5 higher risk of recurrence after PD without a vein resection, respectively. Similarly, stages pT3 and pN2 each increased the risk for LR by approximately three-fold in these patients. The beneficial effect of adjuvant chemotherapy was depicted by a 2.7–2.8 lower risk for any disease recurrence. However, these results (significance of pT, pN and adjuvant chemotherapy) were not replicated for the vein resection subgroup (Table 3 and Table 4). In general, the above findings concur with the published evidence, supporting that a higher pT stage of the disease and regional lymphadenopathy adversely affect oncological outcomes [43]. The loss of significance in the vein resection subgroup may be explained by the small number of patients with early disease stage (pT1 and/or pN0) that resulted in a type I error.

The risk analysis of the resection margins identified R1 surgical margins status as an independent predictor for OS only after PD with vein resection, conferring a 1.9 times risk increase (Table 2). R1 resection, however, independently increased the risk for any disease recurrence (shorter DFS) by 1.7 times for both subgroups, as well as for LR (by 1.8 times after PD without vein resection and by 2.5 times after PD with vein resection) (Table 3 and Table 4). After multivariable analysis of each margin separately, SMA, posterior, BD and pancreatic transection margins were the only ones that independently affected oncological outcomes. More specifically, BD transection margin positivity in patients without a vein resection doubled the risk for a shorter OS and increased the risk for any disease recurrence by 3.4 times and specifically for LR by 5.3 times (Table 2, Table 3 and Table 4). Positivity of the pancreas transection margin increased the risk for a shorter OS by 2.5 times in patients after PD with vein resection (Table 2). The BD is almost invariably transected during PD just proximal to the cystic duct/BD confluence, while the pancreas is transected at the neck of the gland. Despite some evidence that positivity of these margins does not impact OS [44], practice guidelines [45] suggest that these margins are sent intra-operatively for frozen section to avoid positivity in margins that can be further resected. Our study covers a wide period of practice and frozen sections were not the standard of care for a part of this period, which may have resulted in the reported rates of margin positivity. Furthermore, sometimes these margins are deemed positive on permanent fixation of the specimen despite been reported as negative during frozen sections due to the difficulties in pathological assessment of frozen sections for some cases of PDAC [46]. SMA margin positivity conferred a 2.2 times risk increase for any disease recurrence (Table 3), as well as for LR in patients without a vein resection (Table 4). The significance of the SMA margin is well described in the literature [11,47], as are the possible ways to improve or accentuate this [30,31,32]. A positive posterior surface margin conferred a 3.3 times increased risk for recurrence in the vein resection subgroup (Table 3).

Interestingly, the vein-related resection margins, PVG and vein transection margin were not identified as independent predictors for OS, DFS or LR in any group (Table 2, Table 3 and Table 4). On the contrary, vein wall tumour invasion (only possible to assess in patients with vein resection) was an independent predictor of OS (1.7–2 times greater risk) and DFS (1.9–2.2 times greater risk) in all multivariable models (Table 2 and Table 3), while it replaced surgical margin positivity as the only parameter independently predicting LR during multivariable analysis of separate resection margins, increasing the risk by 2.4 times (Table 4). These results concurred with those of other studies [14,22,23,48,49,50,51,52,53], which also reported that there is no difference in OS in patients with superficial venous involvement without vein wall tumour invasion. The lack of true tumour involvement of the vein in 20–43% of resections was offered as a possible explanation [49,50,53,54], which in our study was true in 63% of the vein resection cases. Therefore, poorer oncological outcomes in these cases may be more related to tumour biology, more extensive nodal involvement or peri-operative factors (such as major intra-operative blood loss) [55].

The limitations of our study include its retrospective and single-centre design. Additionally, as the study covered a 10-year period with changes in the preferred systemic treatment regimens for PDAC, treatment selection time bias is inevitable. The study focused on early stages of PDAC as staged with imaging on diagnosis and based on the NCCN criteria. Due to the discordance between radiological and pathological staging [55], a small number of more pathologically advanced tumours (pT3) were also included, and this may have affected the results of the study. Nonetheless, as resectability depends on radiological staging this is a real-life limitation and represents real practice. Finally, our cohort did not include any cases treated with neoadjuvant treatment and therefore any benefit of such an approach, especially in cases with infiltrative tumours, could not be investigated.

## 5. Conclusions

Our results conclude that vein wall tumour invasion may be a more reliable predictor of oncological outcomes compared to traditionally reported parameters such as the PVG and vein transection margin. Future studies should focus on possible pre-operative investigations that could identify these cases and management pathways that could yield a survival benefit, such as the use of neoadjuvant treatments.

## Figures and Tables

**Figure 1 diagnostics-13-03465-f001:**
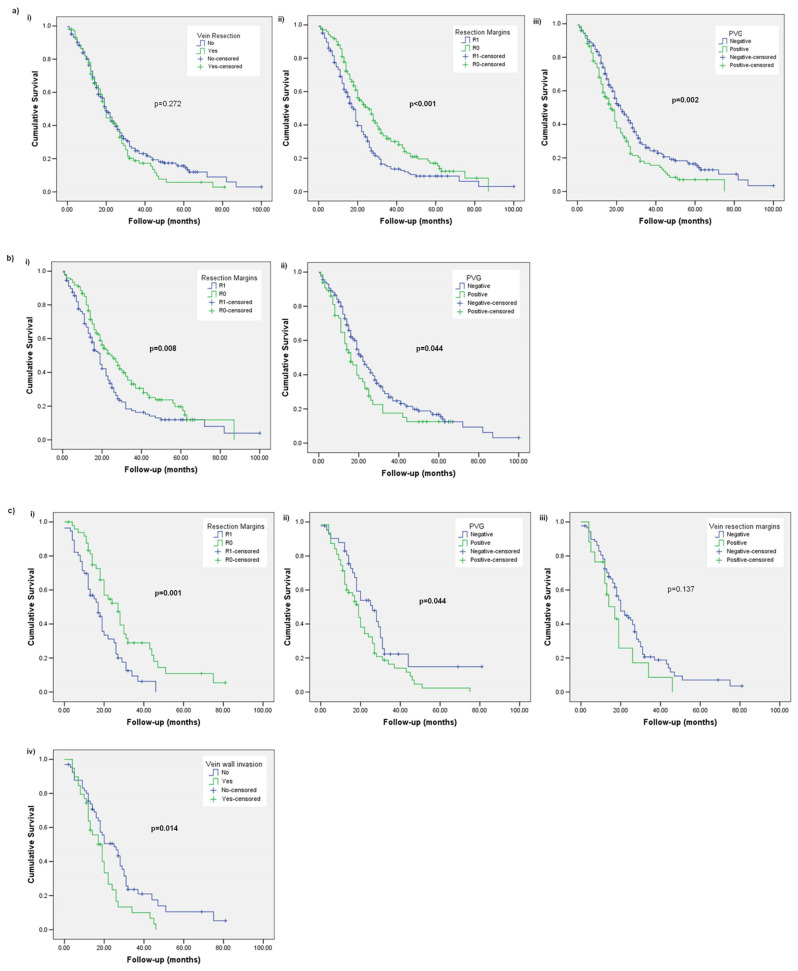
Kaplan–Meier curves comparing overall survival (OS) for (**a**) the whole cohort with (**i**) vein resection or nota, (**ii**) resection margins and (**iii**) portomesenteric vein groove; (**b**) patients without a vein resection with (**i**) resection margins and (**ii**) portomesenteric vein groove; (**c**) patients with a vein resection with (**i**) resection margins, (**ii**) portomesenteric vein groove, (**iii**) vein resection margins and (**iv**) vein wall invasion.

**Figure 2 diagnostics-13-03465-f002:**
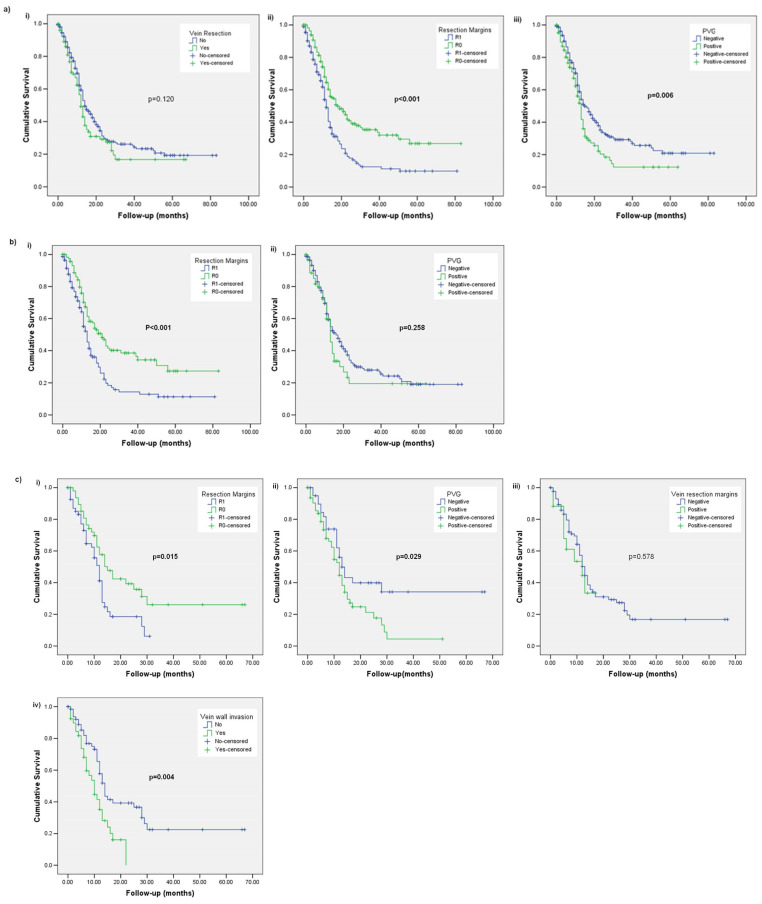
Kaplan–Meier curves comparing overall survival (DFS) for (**a**) the whole cohort with (**i**) vein resection or not, (**ii**) resection margins and (**iii**) portomesenteric vein groove; (**b**) patients without a vein resection with (**i**) resection margins and (**ii**) portomesenteric vein groove; (**c**) patients with a vein resection with (**i**) resection margins, (**ii**) portomesenteric vein groove, (**iii**) vein resection margins and (**iv**) vein wall invasion.

**Figure 3 diagnostics-13-03465-f003:**
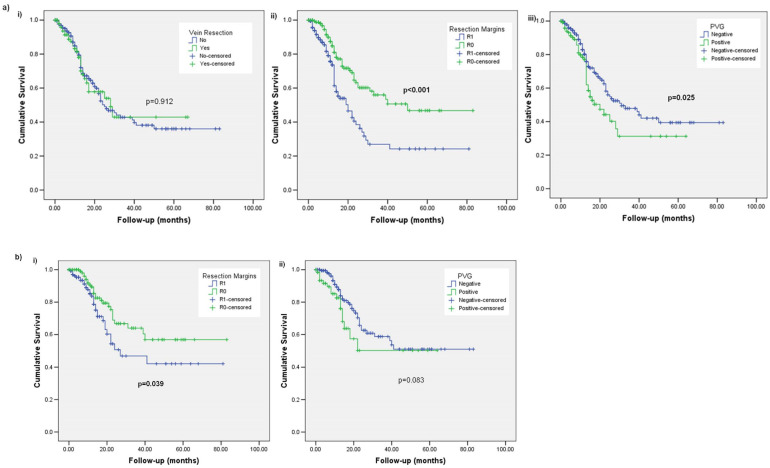
Kaplan–Meier curves comparing local recurrence (LR) for (**a**) the whole cohort with (**i**) vein resection or not, (**ii**) resection margins and (**iii**) portomesenteric vein groove; (**b**) patients without a vein resection with (**i**) resection margins and (**ii**) portomesenteric vein groove; (**c**) patients with a vein resection with (**i**) resection margins, (**ii**) portomesenteric vein groove, (**iii**) vein resection margins and (**iv**) vein wall invasion.

**Table 1 diagnostics-13-03465-t001:** Complete cohort and subgroups characteristics.

	Complete Cohort (*n* = 372)	No Vein Resection (*n* = 267)	Vein Resection (*n* = 105)	*p*
Age (median and range; years)	69 (34–85)	69 (34–85)	71 (48–82)	**0.025**
Sex (male:female)	193:179 (52%:48%)	148:119 (55%:45%)	45:60 (43%:57%)	**0.038**
Pre-operative stage on imaging				**<0.001**
Resectable	279 (75%)	248 (93%)	31 (30%)	
BR-V	93 (25%)	19 (7%)	74 (70%)	
Pre-operative CA19-9 (median and range)	200 (2–36,000)	193 (2–36,000)	237 (2–10,000)	0.929
Vein resection				-
Yes	105 (28%)	N/A	105 (100%)	
No	267 (72%)			
Vein Reconstruction				-
Primary	85 (23%)	N/A	85 (81%)	
Interposition graft	20 (5%)		20 (19%)	
pT				0.430
pT1	35 (9%)	28 (10%)	7 (7%)	
pT2	308 (83%)	218 (82%)	90 (86%)	
pT3	29 (8%)	21 (8%)	8 (7%)	
pN				0.172
pN0	44 (12%)	38 (14%)	6 (5%)	
pN1	185 (39%)	101 (38%)	44 (42%)	
pN2	183 (49%)	128 (48%)	56 (53%)	
Peri-neural invasion (PNI)	344 (93%)	245 (92%)	99 (94%)	0.515
Intra-vascular invasion (VI)	334 (90%)	239 (90%)	95 (91%)	1.000
Surgical margin status				0.818
Negative (R0)	170 (46%)	121 (45%)	45 (46%)	
Positive (R1)	202 (54%)	146 (55%)	53 (54%)	
Specific margin/surface positivity				
Anterior	21 (6%)	17 (6%)	4 (4%)	**0.457**
Posterior	31 (8%)	27 (10%)	4 (4%)	**0.059**
Vein groove	128 (34%)	54 (24%)	63 (60%)	**<0.001**
Vein transection	17 (5%)	-	17 (16%)	**N/A**
SMA	65 (18%)	47 (18%)	18 (17%)	**1.000**
Pancreas	53 (14%)	31 (12%)	22 (21%)	**0.022**
BD	11 (3%)	9 (3%)	2 (2%)	**0.527**
Duodenum	14 (4%)	10 (4%)	3 (3%)	**0.766**
Vein wall invasion	39 (11%)	N/A	39 (37%)	-
Post-operative complications	154 (42%)	108 (41%)	46 (44%)	0.640
Comprehensive Complications Index (median and range)	0 (0–100)	0 (0–100)	0 (0–100)	0.887
Length of stay (median and range; days)	9 (2–200)	9 (2–84)	9 (4–200)	**0.032**
Adjuvant chemotherapy	267 (73%)	193 (73%)	74 (73%)	1.000
Follow-up (median and range; months)	18 (0–100)	17 (0–100)	18 (0–81)	0.444
Disease recurrence	226 (61%)	159 (60%)	67 (64%)	0.639
Local	84 (23%)	60 (23%)	24 (23%)	
Distant	107 (29%)	70 (26%)	37 (35%)	
Both	35 (9%)	29 (11%)	6 (6%)	

BR-V: borderline resectable with venous-only involvement; BD: bile duct, SMA: superior mesenteric artery, N/A: not applicable.

**Table 2 diagnostics-13-03465-t002:** Risk analysis for overall survival (OS).

	Univariable Analysis *p*; HR (95% CI)		
	Complete Cohort	No Vein Resection Subgroup	Vein Resection Subgroup
Age	0.791; 1.002 (0.989–1.015)	0.779; 1.002 (0.987–1.018)	0.886; 0.998 (0.975–1.022)
Sex	0.579; 0.936 (0.740–1.183)	0.539; 0.915 (0.690–1.214)	0.735; 0.927 (0.598–1.437)
Pre-operative CA19–9	0.948; 1.000 (1.000–1.000)	0.915; 1.000 (1.000–1.000)	0.946; 1.000 (1.000–1.000)
Pre-operative stage	0.075; 1.270 (0.976–1.654)	0.745; 0.900 (0.476–1.701)	0.054; 1.671 (0.990–2.818)
Vein resection	0.282; 1.151 (0.891–1.488)	-	-
pT	**0.010; 1.473 (1.097–1.977)**	**0.048; 1.389 (1.004–1.922)**	0.109; 1.808 (0.876–3.732)
pN	**<0.001; 1.679 (1.401–2.013)**	**<0.001; 1.715 (1.389–2.117)**	**0.018; 1.558 (1.078–2.252)**
Surgical margin status	**<0.001; 1.565 (1.234–1.983)**	**0.009; 1454 (1.096–1.929)**	**0.002; 2.041 (1.299–3.206)**
Anterior surface	0.949; 1.017 (0.613–1.685)	0.830; 1.064 (0.605–1.869)	0.872; 0.980 (0.308–3.111)
Posterior surface	0.141; 1.342 (0.907–1.986)	0.210; 1.315 (0.857–2.018)	0.178; 2.003 (0.729–5.503)
Vein groove	**0.002; 1.466 (1.148–1.872)**	**0.049; 1.389 (1.001–1.928)**	**0.050; 1.572 (1.000–2.471)**
Vein transection margin	0.442; 0.949 (0.829–1.085)	N/A	0.148; 1.535 (0.859–2.741)
SMA margin	**0.002; 1.585 (1.181–2.128)**	**0.004; 1.667 (1.175–2.365)**	0.232; 1.398 (0.807–2.421)
Pancreas margin	0.085; 1.366 (0.958–1.947)	0.821; 1.058 (0.648–1729)	**0.011; 1.998 (1.169–3.415)**
BD margin	0.061; 1.829 (0.971–3.444)	**0.046; 2.058 (1.012–4.185)**	0.678; 1.348 (0.329–5.518)
Duodenal/gastric margin	0.997; 1.001 (0.532–1.884)	0.847; 0.928 (0.436–1.976)	0.751; 1.205 (0.379–3.831)
Vein wall invasion	0.730; 0.975 (0.842–1.128)	N/A	**0.017; 1.728 (1.103–2.706)**
PNI	0.844; 0.954 (0.598–1.521)	0.698; 1.114 (0.646–1.920)	**0.024; 0.345 (0.137–0.868)**
VI	0.414; 1.187 (0.787–1.791)	0.626; 1.129 (0.694–1.835)	0.389; 1.408 (0.646–3.065)
Post-operative complications	0.885; 1.018; (0.801–1.293)	0.646; 0.935 (0.701–1.247)	0.238; 1.296 (0.842–1.996)
Comprehensive complications index	**0.001; 1.013 (1.005–1.020)**	**0.005; 1.012 (1.003–1.020)**	**0.032; 1.015 (1.001–1.030)**
Adjuvant chemotherapy	**<0.001; 0.393 (0.304–0.509)**	**<0.001; 0.331 (0.243–0.451)**	**0.030; 0.590 (0.366–0.951)**
	**(A) Multivariable models for total margin status** ** *p* ** **; HR (95% CI)**		
	**Complete cohort**	**No vein resection subgroup**	**Vein resection subgroup**
Pre-operative stage Resectable as reference	NS	Not included as *p* > 0.200 in univariable analysis	0.063; 1.678 (0.972–2.897)
pT pT1 as reference pT2 pT3	**0.018**	NS	**0.009**

	0.921; 1.021 (0.673–1.550)		0.881; 1.076 (0.413–2.802)
	**0.030; 1.863 (1.061–3.270)**		**0.029; 3.663 (1.145–11.723)**
pN pN0 as reference pN1 pN2	**<0.001**	**<0.001**	0.063

	**<0.001; 2.591 (1.595–4.209)**	**<0.001; 4.388 (2.465–7.811)**	0.911; 0.942 (0.330–2.688)
	**<0.001; 3.736 (2.318–6.019)**	**<0.001; 5.817 (3.339–10.137)**	0.340; 1.635 (0.595–4.494)
Surgical margin status R0 as reference	**0.015; 1.351 (1.059–1.723)**	NS	**0.009; 1.942 (1.181–3.195)**
Vein wall invasion Negative as reference	Not included as *p* > 0.200 in univariable analysis	Not included as *p* > 0.200 in univariable analysis	**0.031; 1.708 (1.051–2.775)**
PNI Negative as reference	Not included as *p* > 0.200 in univariable analysis	Not included as *p* > 0.200 in univariable analysis	**0.022; 0.318 (0.119–0.851)**
Adjuvant chemotherapy No chemotherapy as reference	**<0.001; 2.909 (2.235–3.787)**	**<0.001; 4.571 (3.260–6.408)**	NS
	**(B) Multivariable models for individual surgical margins** ** *p* ** **; HR (95% CI)**		
	**Complete cohort**	**No vein resection subgroup**	**Vein resection subgroup**
pT pT1 as reference pT2 pT3	**0.009**	NS	**0.002**

	0.813; 1.052 (0.691–1.601)		0.390; 0.667 (0.265–1.680)
	**0.015; 2.030 (1.145–3.599)**		0.084; 2.695 (0.874–8.308)
pN pN0 as reference pN1 pN2	**<0.001**	**<0.001**	NS

	**<0.001; 2.505 (1.542–4.068)**	**<0.001; 4.384 (2.463–7.801)**	
	**<0.001; 3.695 (2.298–5.941)**	**<0.001; 5.712 (3.274–9.966)**	
PVG Negative as reference	0.055; 1.280 (0.995–1.646)	NS	NS
BD margin Negative as reference	NS	**0.048; 2.052 (1.005–4.189)**	Not included as *p* > 0.200 in univariable analysis
Pancreas margin Negative as reference	NS	Not included	**0.001; 2.465 (1.415–4.293)**
Vein wall invasion Negative as reference	Not included as *p* > 0.200 in univariable analysis	N/A	**0.005; 1.947 (1.228–3.089)**
Comprehensive complications index	0.084; 1.007 (0.999–1.015)	NS	NS
Adjuvant chemotherapy No chemotherapy as reference	**<0.001; 2.672 (2.027–3.523)**	**<0.001; 4.639 (3.308–6.507)**	NS

All parameters with *p* < 0.200 in univariable analysis were entered in the multivariable model. Separate models were computed for (A) combined margin status and (B) individual surgical margins. BD: bile duct, SMA: superior mesenteric artery, PNI: perineural invasion, VI: intra-vascular invasion, PVG: portomesenteric vein groove, NS: not significant, N/A: not applicable.

**Table 3 diagnostics-13-03465-t003:** Risk analysis for disease-free survival (DFS).

	Univariable Analysis *p*; HR (95% CI)		
	Complete Cohort	No Vein Resection Subgroup	Vein Resection Subgroup
Age	0.449; 0.994 (0.980–1.009)	0.773; 0.997 (0.980–1.015)	0.189; 0.983 (0.957–1.009)
Sex	0.380; 0.889 (0.684–1.155)	0.391; 0.871 (0.636–1.194)	0.653; 0.895 (0.553–1.449)
Pre-operative CA19–9	0.515; 1.000 (1.000–1.000)	0.438; 1.000 (1.000–1.000)	0.990; 1.000 (1.000–1.000)
Pre-operative stage	**0.010; 1.467 (1.095–1.966)**	0.294; 1.390 (0.751–2.572)	0.110; 1.600 (0.900–2.846)
Vein resection	0.130; 1.248 (0.937–1.661)	-	-
pT	**0.007; 1.607 (1.142–2.262)**	**0.003; 1.780 (1.215–2.608)**	0.965; 1.019 (0.447–2.323)
pN	**<0.001; 1.647 (1.348–2.012)**	**<0.001; 1.751 (1.389–2.208)**	0.227; 1.286 (0.855–1.933)
Surgical margin status	**<0.001; 1.813 (1.388–2.370)**	**<0.001; 1.807 (1.313–2.485)**	**0.019; 1.805 (1.101–2.957)**
Anterior surface	0.809; 0.931 (0.520–1.666)	0.660; 0.860 (0.439–1.685)	0.498; 1.497 (0.466–4.812)
Posterior surface	0.086; 1.470 (0.947–2.284)	0.213; 1.366 (0.836–2.233)	**0.030; 3.127 (1.118–8.748)**
PVG	**0.008; 1.447 (1.101–1.903)**	0.270; 1.231 (0.850–1.783)	**0.036; 1.734 (1.037–2.901)**
Vein transection margin	0.180; 0.903 (0.778–1.048)	N/A	0.589; 1.216 (0.598–2.474)
SMA margin	**<0.001; 1.932 (1.397–2.674)**	**<0.001; 2.384 (1.633–3.480)**	0.656; 1.159 (0.605–2.219)
Pancreas margin	0.445; 1.168 (0.785–1.738)	0.793; 1.074 (0.630–1.831)	0.619; 1.168 (0.634–2.151)
BD margin	0.052; 2.023 (0.995–4.113)	**0.004; 3.047 (1.414–6.568)**	0.549; 0.546 (0.076–3.943)
Duodenal/gastric margin	0.378; 1.330 (0.705–2.508)	0.670; 1.179 (0.552–2.518)	0.288; 1.883 (0.586–6.049)
Vein wall invasion	0.597; 0.957 (0.813–1.126)	N/A	**0.006; 2.019 (1.219–3.345)**
PNI	0.101; 1.700 (0.901–3.207)	0.106; 1.799 (0.882–3.667)	0.921; 1.074 (0.262–4.399)
VI	0.063; 1.563 (0.976–2.503)	0.236; 1.394 (0.805–2.415)	0.125; 2.051 (0.820–5.130)
Post-operative complications	0.152; 0.820 (0.624–1.076)	0.280; 0.836 (0.604–1.157)	0.269; 0.754 (0.458–1.244)
Comprehensive complications index	0.145; 0.993 (0.983–1.002)	0.245; 0.993 (0.982–1.005)	0.300; 0.991 (0.974–1.008)
Adjuvant chemotherapy	**0.003; 0.613 (0.445–0.846)**	**0.012; 0.604 (0.408–0.894)**	0.130; 0.646 (0.367–1.138)
	**(A) Multivariable models for total margin status** ** *p* ** **; HR (95% CI)**		
	**Complete cohort**	**No vein resection subgroup**	**Vein resection subgroup**
Pre-operative stage Resectable as reference	**0.027; 1.398 (1.039–1.883)**	Not included as *p* > 0.200 on univariable analysis	NS
pT pT1 as reference pT2 pT3	**0.014**	**0.015**	Not included as *p* > 0.200 on univariable analysis

	0.925; 0.977 (0.598–1.595)	0.648; 1.144 (0.642–2.038)	
	**0.040; 1.966 (1.030–3.753)**	**0.019; 2.444 (1.160–5.146)**	
pN pN0 as reference pN1 pN2	**<0.001**	**<0.001**	Not included as *p* > 0.200 in univariable analysis

	**<0.001; 2.964 (1.708–5.144)**	**<0.001; 3.450 (1.860–6.399)**	
	**<0.001; 3.970 (2.289–6.885)**	**<0.001; 4.536 (2.459–8.367)**	
Surgical margin status R0 as reference	**<0.001; 1.734 (1.321–2.276)**	**0.002; 1.693 (1.218–2.352)**	**0.044; 1.684 (1.013–2.799)**
Vein wall invasion Negative as reference	Not included as *p* > 0.200 on univariable analysis	N/A	**0.016; 1.894 (1.127–3.182)**
Post-operative complications Negative as reference	**0.015; 0.703 (0.530–0.933)**	Not included as *p* > 0.200 in univariable analysis	Not included as *p* > 0.200 in univariable analysis
Adjuvant chemotherapy No chemotherapy as reference	**<0.001; 2.167 (1.532–3.065)**	**<0.001; 2.653 (1.728–4.075)**	NS
	**(B) Multivariable models for individual surgical margins** ** *p* ** **; HR (95% CI)**		
	**Complete cohort**	**No vein resection subgroup**	**Vein resection subgroup**
Pre-operative stage Resectable as reference	**0.011; 1.476 (1.092–1.993)**	Not included as *p* < 0.200 in univariable analysis	NS
pT pT0 as reference pT1 pT2	**0.038**	**0.038**	Not included as *p* > 0.200 in univariable analysis

	0.680; 0.901 (0.550–1.476)	0.776; 1.088 (0.608–1.949)	
	0.126; 1.672 (0.865–3.234)	**0.047; 2.145 (1.009–4.558)**	
pN pN0 as reference pN1 pN2	**<0.001**	**<0.001**	Not included as *p* > 0.200 in univariable analysis

	**<0.001; 2.887 (1.665–5.006)**	**<0.001; 3.339 (1.795–6.209)**	
	**<0.001; 3.898 (2.248–6.760)**	**<0.001; 4.371 (2.346–8.144)**	
Posterior margin Negative as reference	0.051; 1.585 (0.998–2.516)	Not included as *p* > 0.200 in univariable analysis	**0.024; 3.295 (1.173–9.254)**
SMA margin Negative as reference	**0.002; 1.688 (1.208–2.357)**	**<0.001; 2.197 (1.480–3.262)**	Not included as *p* > 0.200 in univariable analysis
BD margin Negative as reference	NS	**0.002; 3.441 (1.580–7.494)**	Not included as *p* > 0.200 in univariable analysis
Vein wall invasion Negative as reference	Not included as *p* > 0.200 in univariable analysis	N/A	**0.003; 2.169 (1.306–3.602)**
Post-operative complications Negative as reference	**0.009; 0.682 (0.513–0.907)**	Not included as *p* > 0.200 in univariable analysis	Not included as *p* > 0.200 in univariable analysis
Adjuvant chemotherapy No chemotherapy as reference	**<0.001; 2.127 (1.508–3.001)**	**<0.001; 2.827 (1.841–4.340)**	NS

All parameters with *p* < 0.200 in univariable analysis were entered in the multivariable model. Separate models were computed for (A) combined margin status and (B) individual surgical margins. BD: bile duct, SMA: superior mesenteric artery, PNI: perineural invasion, VI: intra-vascular invasion, PVG: portomesenteric vein groove, NS: not significant, N/A: not applicable.

**Table 4 diagnostics-13-03465-t004:** Risk analysis for local recurrence (LR).

	Univariable Analysis *p*; HR (95% CI)		
	Complete Cohort	No Vein Resection Subgroup	Vein Resection Subgroup
Age	0.844; 1.002 (0.981–1.023)	0.966; 1.001 (0.976–1.025)	0.800; 1.006 (0.963–1.050)
Sex	0.668; 1.083 (0.751–1.562)	0.525; 1.147 (0.751–1.752)	0.875; 0.943 (0.453–1.963)
Pre-operative CA19–9	0.243; 1.000 (1.000–1.000)	0.322; 1.000 (1.000–1.000)	0.705; 1.000 (1.000–1.000)
Pre-operative stage	0.266; 1.276 (0.831–1.958)	0.668; 1.219 (0.492–3.021)	0.206; 1.787 (0.726–4.399)
Vein resection	0.914; 1.024 (0.671–1.562)	N/A	N/A
pT	**0.016; 1.820 (1.121–2.957)**	**0.023; 1.839 (1.088–3.108)**	0.337; 1.871 (0.521–6.716)
pN	**0.000; 1.673 (1.269–2.206)**	**<0.001; 1.856 (1.357–2.540)**	0.739; 1.108 (0.607–2.022)
Surgical margin status	**<0.001; 2.135 (1.464–3.115)**	**0.002; 1.997 (1.294–3.083)**	**0.022; 2.452 (1.140–5.273)**
Anterior surface	0.705; 1.159 (0.540–2.492)	0.743; 1.149 (0.501–2.636)	0.810; 1.279 (0.171–9.557)
Posterior surface	0.064; 1.762 (0.968–3.207)	0.090; 1.731 (0.918–3.261)	0.445; 2.201 (0.290–16.702)
PVG	**0.028; 1.538 (1.048–2.257)**	0.079; 1.540 (0.952–2.492)	0.199; 1.659 (0.767–3.591)
Vein transection margin	0.978; 1.003 (0.805–1.250)	N/A	0.603; 1.328 (0.456–3.872)
SMA margin	**0.004; 1.997 (1.248–3.197)**	**0.001; 2.432 (1.413–4.185)**	0.646; 1.255 (0.476–3.314)
Pancreas margin	0.203; 1.412 (0.830–2.404)	0.305; 1.414 (0.729–2.741)	0.529; 1.339 (0.539–3.326)
BD margin	**0.003; 3.588 (1.561–8.248)**	**0.001; 5.117 (2.028–12.913)**	0.798; 1.298 (0.176–9.605)
Duodenal/gastric margin	0.281; 1.571 (0.690–3.576)	0.693; 1.224 (0.448–3.342)	0.111; 3.277 (0.762–14.094)
Vein wall invasion	0.552; 1.076 (0.845–1.371)	N/A	**0.030; 2.365 (1.089–5.133)**
PNI	**0.030; 4.697 (1.160–19.030)**	**0.043; 4.246 (1.043–17.292)**	0.517; 21.287 (0.002–222017.1)
VI	0.101; 1.769 (0.895–3.497)	0.260; 1.560 (0.720–3.383)	0.223; 2.454 (0.579–10.403)
Post-operative complications	0.173; 0.766 (0.523–1.124)	0.259; 0.775 (0.497–1.207)	0.356; 0.697 (0.324–1.500)
Comprehensive complications index	0.056; 0.986 (0.971–1.000)	0.075; 0.984 (0.967–1.002)	0.390; 0.988 (0.962–1.015)
Adjuvant chemotherapy	0.388; 1.244 (0.758–2.040)	0.706; 0.889 (0.481–1.642)	0.324; 1.538 (0.654–3.618)
	**(A) Multivariable models for total margin status** ** *p* ** **; HR (95% CI)**		
	**Complete cohort**	**No vein resection subgroup**	**Vein resection subgroup**
pT pT1 as reference pT2 pT3	**0.005**	**0.027**	Not included as *p* > 0.200 in univariable analysis

	0.993; 1.003 (0.518–1.941)	0.719; 1.146 (0.547–2.399)	
	**0.020; 2.756 (1.174–6.473)**	**0.028; 2.949 (1.121–7.755)**	
pN pN0 as reference pN1 pN2	**0.014**	**0.003**	Not included as *p* > 0.200 in univariable analysis

	0.112; 1.728 (0.880–3.391)	0.114; 1.794 (0.8168–3.707)	
	**0.007; 2.516 (1.288–4.916)**	**0.002; 3.082 (1.517–6.259)**	
Surgical margin statusR0 as reference	**<0.001; 2.041 (1.391–2.996)**	**0.007; 1.845 (1.183–2.877)**	**0.022; 2.452 (1.140–5.273)**
	**(B) Multivariable models for individual surgical margins** ** *p* ** **; HR (95% CI)**		
	**Complete cohort**	**No vein resection subgroup**	**Vein resection subgroup**
pT pT1 as reference pT2 pT3	**0.010**	NS	Not included as *p* > 0.200 in univariable analysis

	0.826; 0.928 (0.476–1.809)		
	**0.045; 2.398 (1.020–5.638)**		
pN pN0 as reference pN1 pN2	**0.014**	**0.004**	Not included as *p* > 0.200 in univariable analysis

	0.133; 1.684 (0.853–3.326)	0.134; 1.741 (0.843–3.597)	
	**0.008; 2.498 (1.269–4.917)**	**0.003; 2.981 (1.466–6.060)**	
SMA margin Negative as reference	**0.012; 1.854 (1.145–3.001)**	**0.006; 2.172 (1.245–3.788)**	Not included as *p* > 0.200 in univariable analysis
BD margin Negative as reference	**0.005; 3.342 (1.431–7.808)**	**0.001; 5.276 (2.067–13.468)**	Not included as *p* > 0.200 in univariable analysis
PNI Negative as reference	0.99; 3.277 (0.799–13.441)	NS	Not included as *p* > 0.200 in univariable analysis
Vein wall invasion Negative as reference	Not included as *p* > 0.200 in univariable analysis	N/A	**0.030; 2.365 (1.089–5.133)**

All parameters with *p* < 0.200 in univariable analysis were entered in the multivariable model. Separate models were computed for (A) combined margin status and (B) individual surgical margins. BD: bile duct, SMA: superior mesenteric artery, PNI: perineural invasion, VI: intra-vascular invasion, PVG: portomesenteric vein groove, NS: not significant, N/A: not applicable.

## Data Availability

Data are available through the corresponding author on request.

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
