# Peer review of "Vein Wall Invasion Is a More Reliable Predictor of Oncological Outcomes than Vein-Related Margins after Pancreaticoduodenectomy for Early Stages of Pancreatic Ductal Adenocarcinoma"

_diagnostics, 2023, doi:10.3390/diagnostics13223465_

Round 1

Reviewer 1 Report

Comments and Suggestions for Authors

Thank you for the opportunity to review this important paper:

- I would suggest citing in the introduction: 32889866, 34791069, 29191690, 26663252; 28833055

- results are too long, consider adding supplemental material (analysis DFS and LR?)

- start discussion with alinea summarizing most important findings 

- i dont agree with lines 289-291: the PVG is important in venous resection, the extent of venous resection not always as long as the length of the PVG?

Author Response

Dear Reviewer

Thank you for taking the time and effort to review our manuscript.

  • suggested papers have been sited in the introduction
  • the result section is only about 2 pages long. The additional information is on tables and figures. We understand that there is no limitation from the journal instructions. If there is, then some of the tables and/or figures can be submitted as supplementary material. 
  • the study has several important findings which are analysed in the discussion section. The conclusion is clearly offered in the last paragraph. 
  • the authors appreciate the personal opinion expressed. It is in the same line as the Royal College of Pathologists, but differrent than the American College for example as explained in the manuscript. This difference in opinion and reporting is the reason for conducting this study. Both opinions have been stated and published evidence on either also analysed. The outcomes of our cohort have been also presented and analysed. Based on these, vein wall invasion is a more reliable predictor of oncological outcomes.  

Reviewer 2 Report

Comments and Suggestions for Authors

You forgot to introduce in Figure 22 Kaplan  Meier curves for point (c) so we could not have a overall picture of the DFS.

You forgot to individualize the ”Conclusions” Chapter (5. Conclusions put before the text)

When I look to  the existing Kaplan Meier curves they almost push me to ask you a question: If you want to prove the usefulness of the assessment of vein wall invasion why did you take into  consideration the cases without vein resection? The vein wall  invasion could and should be taken into consideration only in cases with vein resection. The title states a comparison between ”vein wall invasion” and ”vein related margins involvement”.

So, why did you introduce in the assessment the cases without vein resection?

Otherwise, ”Resection Margins” seems to be useful in both situations: without and with vein resection and a better indicator even than ”vein wall invasion” in the group with vein resection.

I feel that you have to make it clear both the purpose, the way you prove it the results and the  conclusion.

Your conclusion is related with the title (which seems to be the purpose) but the study and the results presented (however, very well done) seem to cover a larger area than the purpose proposed.

Think about it and try to refine the huge amount of work you presented and to make it fit with the purpose.

Author Response

Dear reviewer

Thank you for taking the time and effort to review our paper.

  • KM curve added
  • headline for conclusions added
  • the aim of the study (as stated in the last paragraph of the introduction) was to assess the oncological prognostic significance of the different pathological variables of venous involvement in patients undergoing PD for early stage PDAC. This is fully described by the 3 pathological variables: vein groove margin, vein resection margin and vein wall invasion. The latter two are only relevant if a vein resection is performed. Hence a subgroup analysis was performed for both univariate and multivariate analysis. There was no intention to compare these pathological variables, as more than one could have been proven to be significant (which was not the case). The traditionally reported vein related margins were not indetified as predictors for any oncological outcome in any subgroup. On the contrary, the vein wall invasion was. Furthermore, focusing only in the vein resection subgroup, would have excluded more than two thrids of patients (that did not require vein resection) with possible "venous involvement" on the vein groove margin that could have proved to be significant for their oncological outcomes. For this subgroup though, other pathological parameters (rather than the vein groove margin positivity) were identified as independent predictors.